# Understanding symptom appraisal and help-seeking in people with symptoms suggestive of pancreatic cancer: a qualitative study

Katie Mills,[1] Linda Birt,[1] Jon D Emery,[2] Nicola Hall,[3] Jonathan Banks,[4] Margaret Johnson,[1] John Lancaster,[1] William Hamilton,[5] Greg P Rubin,[3] Fiona M Walter[1]

[1]Department of Public Health and Primary Care, The Primary Care Unit, University of Cambridge, Cambridge, UK
[2]Department of General Practice, Primary Care Cancer Research, University of Melbourne, Carlton, Victoria, Australia
[3]Evaluation Research Development Unit, School of Medicine, Pharmacy & Health, Durham University, Bristol, UK
[4]Centre for Academic Primary Care, School of Social and Community Medicine, University of Bristol, Bristol, UK
[5]Department of Primary Care Diagnostics, College House, St Luke's Campus, University of Exeter, Exeter, UK

**Correspondence to**
Dr Katie Mills;
ko298@medschl.cam.ac.uk

## ABSTRACT

**Objective** Pancreatic cancer has poor survival rates due to non-specific symptoms leading to later diagnosis. Understanding how patients interpret their symptoms could inform approaches to earlier diagnosis. This study sought to explore symptom appraisal and help-seeking among patients referred to secondary care for symptoms suggestive of pancreatic cancer.

**Design** Qualitative analysis of semistructured in-depth interviews. Data were analysed iteratively and thematically, informed by the Model of Pathways to Treatment.

**Participants and setting** Pancreatic cancer occurs rarely in younger adults, therefore patients aged ≥40 years were recruited from nine hospitals after being referred to hospital with symptoms suggestive of pancreatic cancer; all were participants in a cohort study. Interviews were conducted soon after referral, and where possible, before diagnosis.

**Results** Twenty-six interviews were conducted (cancer n=13 (pancreas n=9, other intra-abdominal n=4), non-cancer conditions n=13; age range 48–84 years; 14 women). Time from first symptoms to first presentation to healthcare ranged from 1 day to 270 days, median 21 days. We identified three main themes. Initial symptom appraisal usually began with intermittent, non-specific symptoms such as tiredness or appetite changes, attributed to diet and lifestyle, existing gastrointestinal conditions or side effects of medication. Responses to initial symptom appraisal included changes in meal type or frequency, or self-medication. Symptom changes such as alterations in appetite and enjoyment of food or weight loss usually prompted further appraisal. Triggers to seek help included a change or worsening of symptoms, particularly pain, which was often a 'tipping point'. Help-seeking was often encouraged by others. We found no differences in symptom appraisal and help-seeking between people diagnosed with cancer and those with other conditions.

**Conclusions** Greater public and healthcare professional awareness of the combinations of subtle and intermittent symptoms, and their evolving nature, is needed to prompt timelier help-seeking and investigation among people with symptoms of pancreatic cancer.

### Strengths and limitations of this study

► We believe this is the first study to compare the symptom appraisal and help-seeking experiences of patients referred to secondary care with symptoms suggestive of pancreatic cancer between people subsequently diagnosed with pancreatic cancer and people diagnosed with other non-cancer conditions. We were unable to identify any differences.

► The data collection and analysis of this study were guided by the model of pathways to treatment as recommended by the Aarhus statement to improve the design and reporting of studies on early cancer diagnosis.

► Risks of recall bias and post hoc rationalisation were reduced by recruiting at the time of referral to specialist care, and interviewing patients before or close to diagnosis.

► The subtlety of initial symptoms and their evolution over time is a novel finding, and provides some understanding of the complexities faced by patients when appraising their symptoms and seeking help in a timely way.

► Some of the most seriously ill patients were unable to be interviewed and their experiences may have differed from those in our sample.

## INTRODUCTION

Pancreatic cancer is the ninth most common cancer in the UK, with about 8000 cases diagnosed every year. Only 22% men and 20% women currently survive pancreatic cancer for 1 year or more, and it has the poorest 5-year survival rates of all cancers at less than 4%.[1] This poor prognosis is mainly due to patients being diagnosed at a stage when curative treatment is not possible. Survival could be improved if patients could be diagnosed earlier.[2 3]

The key symptoms of pancreatic cancer have primarily been described from studies of patients after a diagnosis has been made. Prediagnosis case-control studies from the USA[4] and UK[5 6] confirm that these symptoms

are also relevant for the community when making healthcare management decisions. While jaundice is the only symptom which is strongly predictive of pancreatic cancer, a number of non-specific symptoms such as nausea, vomiting, abdominal pain, back pain, constipation, diarrhoea, weight loss and malaise are also weakly associated.[6] The non-specific nature of these symptoms creates challenges for early recognition of a potentially serious condition by both patients and their General Practitioners (GP). In a large English database study, over 40% of patients diagnosed with pancreatic cancer had visited their GP at least three times before referral to specialist care.[7]

There is limited evidence about the evolution of symptoms of pancreatic cancer over time, and how patients appraise their symptoms and decide to seek medical advice. One of the few studies to date interviewed patients or their relatives many months after diagnosis;[8] such studies may be biased by post hoc rationalisation and recall bias. Furthermore, many symptoms of pancreatic cancer are also symptoms of other gastrointestinal or intra-abdominal cancers such as gall bladder, colon and ovarian cancer, which may be equally difficult to diagnose early.[7]

The aim of this study was therefore to use qualitative approaches to gain understanding of barriers and facilitators to symptom appraisal and help-seeking decisions among patients with symptoms suggestive of pancreatic cancer much earlier in their diagnostic pathway, to contribute to the development of interventions to promote earlier or more timely cancer diagnosis. Similar studies have been conducted for people with symptoms suggestive of lung and colorectal cancers.[9–11] We recruited patients who were newly referred to hospital, thus providing the opportunity to investigate the complexities of patients' symptom appraisal and decision-making, and explore pathways from their first symptom to first presentation in primary care, emergency presentations and referrals to specialist services.

## METHODS
### Setting
This in-depth, face-to-face interview study was nested within the SYMPTOM Pancreas Study, a prospective cohort study investigating associations between symptoms and other factors on the total diagnostic interval and stage of diagnosis among patients with symptoms suggestive of pancreatic cancer.[12] The patients were recruited from nine hospitals in two regions of England (North-East and Eastern).

### Recruitment
Patients were recruited when referred to hospital via routine or urgent (2-week wait) routes to a gastroenterology or hepatopancreatobiliary clinic or relevant ultrasound department, or admitted via Accident and Emergency (A&E) or other specialists. Pancreatic cancer occurs rarely in younger adults, therefore patients were eligible if they were aged 40 years and over, and their GPs had reported symptoms suggestive of pancreatic cancer. They were initially sent an invitation letter and SYMPTOM Study questionnaire by post; people who completed the questionnaire were able to indicate whether they would be willing to take part in an interview. We purposively sampled patients (by region, age, gender, diagnosis) to obtain participants with a range of demographic and diagnosis characteristics.

### Data collection
Interviews were undertaken between March 2012 and March 2014 by KM, LB and NH, all of who have extensive qualitative research experience. They took place as soon as possible after referral to hospital. All diagnoses were confirmed from review of secondary care medical records.

Before starting the interview, the researcher explained the research process and written consent was obtained from the participant. The interview was semistructured, with the schedule (online supplementary file 1) developed from similar interviews undertaken with people recently diagnosed with cancer.[13] By using open-ended questions we were able to explore the patients' appraisal of their symptoms and decisions to seek help. We encouraged the interviewees to share how they made sense of their symptoms and why they chose to seek help in the context of previous experiences and existing medical conditions. To assist in the clarification of the sequences of events and key dates along the pathway to diagnosis, we used a specifically designed calendar-landmarking instrument.[14]

The interviews lasted between 40 min and 70 min and were conducted in participants' homes. A relative, usually a spouse, was also present at several interviews at the participant's request: they sometimes assisted in recall of events and confirmed participants' accounts, but were not consented, and only the participant's words were included as data in analyses. Interviews continued until no new themes were identified in three consecutive interviews and data saturation was reached.[15] All interviews were audio-recorded, then professionally transcribed verbatim and anonymised via a confidential service.

### Analyses
#### Theoretical model
Both data collection and analysis were underpinned by the theoretical approach of the Model of Pathways to Treatment[16 17] as recommended by the Aarhus statement to improve the design and reporting of studies on early cancer diagnosis.[18] The model considers the contributions of disease, patient and healthcare system factors to four intervals: appraisal, help-seeking, diagnostic and pretreatment. The first event is the detection of initial bodily changes, described as the 'time point when a person becomes aware of body sensations or visual alterations, regardless of the meaning assigned to the change', these then develop over time to 'symptoms', defined as a bodily

change perceived to be abnormal. Symptom appraisal leads to perceiving a reason to seek help, and consulting with a healthcare professional. These key events represent the 'time to presentation' (TTP) period. We defined TTP as the interval between the patient-reported date of first noticing a symptom and their first consultation with a healthcare professional.

## Analytical processes

The analytical process was iterative, starting after the first few interviews. Three researchers with social sciences expertise (KM, LB, NH) worked together on an initial thematic analysis with inductive coding,[19] and then adopted a deductive approach guided by the framework of the Model of Pathways to Treatment[16 17 20] to construct themes and subthemes. These approaches were combined to ensure a rigorous and systematic progression through the five analytical steps: familiarisation with data; developing a thematic framework; indexing data to framework; mapping and questioning the data; and theoretical interpretation.[21] The model was used at several points in the analysis process: first, to provide structure to understanding the patient pathways to diagnosis; second, to facilitate comparisons between participants diagnosed with cancer and those who presented with similar symptoms but were diagnosed with non-cancer conditions. Finally, the researchers examined patient data by characteristics, including age, gender, comorbidity, geographical region and diagnosis, to seek patterns and non-confirming cases.

The study patient representatives (MJ, JL) have personal experiences of cancer, which was reflected in their interpretation of the data. They contributed to all stages of analysis by reading transcripts and regular meetings with the research team, and the other authors (all with clinical or social science expertise) contributed to the interpretation of the data. Data management was assisted by NVivo V.9.

## RESULTS
## Sample (table 1)

Of the participants, 302/407 (74%) recruited into the main SYMPTOM Pancreas Study expressed an interest in an interview. Thirty-four individuals were recruited following purposive sampling, but 8 were not interviewed as they were too unwell to participate. Six were unaware of their diagnosis at interview.

Half (13/26) of the interviewees were ultimately diagnosed with cancer, 9 with pancreatic cancer (3 early stage, 3 late stage, 3 unstaged) and 4 with other cancers (duodenal, gall bladder, oesophageal, lymphoma). The 13 participants with non-cancer diagnoses included hiatus hernia, gallstones and pancreatitis. All participants were interviewed soon after their referral to specialist care (range: 1 day to 18 weeks), with a quarter (6, 23%) prior to diagnosis, half (12, 46%) within 4 weeks of diagnosis, and the remainder within 18 weeks (8, 31%).

**Table 1** Participant characteristics

| | Cancer diagnosis n=13 (%) | Other conditions diagnosis n=13 (%) |
|---|---|---|
| English region | | 9 (69) |
| Eastern | 10 (77) | |
| North-Eastern | 3 (23) | 4 (31) |
| Gender | | |
| Female | 6 (46) | 8 (62) |
| Male | 7 (54) | 5 (38) |
| Age | | |
| Mean years (range) | 67 (50–84) | 63 (49–84) |
| 40–59 | 4 (31) | 3 (23) |
| 60–69 | 4 (31) | 4 (31) |
| 70–79 | 2 (15) | 4 (31) |
| 80+ | 3 (23) | 2 (15) |
| Ethnicity | | |
| White British | 13 (100) | 13 (100) |
| Educational qualifications | | |
| Higher education | 3 (23) | 1 (8) |
| Further education/secondary school | 6 (46) | 8 (62) |
| None | 4 (31) | 4 (31) |
| Index of Multiple Deprivation (IMD) quintiles | | |
| Least deprived 1 | 4 (31) | 2 (15) |
| 2 | 4 (31) | 4 (31) |
| 3 | 2 (15) | 3 (23) |
| 4 | 3 (23) | 2 (15) |
| Most deprived 5 | 0 (0) | 2 (15) |
| Diagnoses of cancer | | |
| Pancreas | 9 (69) | – |
| Duodenal | 1 (8) | – |
| Gall bladder | 1 (8) | – |
| Oesophageal | 1 (8) | – |
| Lymphoma | 1 (8) | – |
| Other conditions | | |
| Hiatus hernia | – | 3 (23) |
| Gallstones | – | 2 (15) |
| Pancreatitis | – | 2 (15) |
| Others | – | 6 (46) |
| GP appointments made | | |
| 0* | 0 (0) | 2 (15) |
| 1 | 9 (62) | 6 (46) |
| 2 | 1 (8) | 4 (31) |
| *Emergency presentation 3+ | 3 (23) | 1 (8) |

## Duration of symptoms

Most participants were able to recall when they first noticed their symptom/s and the date they went to their GP, though for some it was difficult to recall the exact date they decided to seek help. The TTP ranged from 1 day to 270 days, with a median of 21 days. Three individuals were frequent attenders to their GP and so we were unable to define the precise first date of presentation. Two-thirds (68%) consulted their GP within 6 weeks of their trigger symptom. While more than half (58%) consulted their GPs only once before referral to secondary care, four patients (16%) saw their GP on more than three occasions.

## Qualitative themes

We were unable to find any clear differences regarding symptom appraisal or help-seeking between the accounts of people diagnosed with cancer and those with other conditions. We therefore report the findings together. We were also unable to find any clear differences according to whether or not participants knew their diagnosis at the time of the interview, or their characteristics including gender, age and region. Extracts from interviews illustrate the findings: each quotation is contextualised, by the patient's gender (M/F), age (40–59 years, 60–69 years, 70–79 years, 80+ years), TTP (weeks) and diagnosis (cancer, non-cancer).

### Theme 1: initial appraisal of symptoms

Two main subthemes emerged as participants described how they first noticed and initially interpreted their symptoms, and often attributed them to a common non-serious cause or condition.

#### Bodily changes

People mainly described initial subtle or non-specific bodily changes such as feeling tired or 'different'. These changes were often intermittent and insidious in their development.

*'I've had this sort of tiredness and a little bit of discomfort on and off for a time'* (M, 70–79 years, 13–26 weeks, cancer)

*'I've felt a bit like as if I've got a nervous tummy, like it gurgles up…'* (F 50–59 years, regular attender to GP, non-cancer)

People also described subtle changes in appetite: *'Well I went off my food a bit I think, my daughter said I went off my food a bit'* (F, 80+ years, 2–4 weeks, cancer)

### Symptom attributions (table 2)

Initial explanations for these bodily changes or symptoms mainly related to diet and other daily activities. Nine participants with other conditions, particularly gastrointestinal conditions such as gastro-oesophageal reflux disease, often appraised their new symptoms against their 'usual' symptoms, and whether their treatment was still effective. This group tended to downplay their symptoms. Some people also attributed these bodily changes or symptoms to other medical conditions or side effects of their medication. Others 'normalised' their initial symptoms to being caused by ageing, common conditions such as a virus or common cold, indigestion or a urine infection.

These explanations were described in the rich context of people's lives, such as being busy, caring for others or the time of the year. When considered in these contexts, most people initially felt that their symptoms were acceptable and did not warrant seeking help:

*'I thought I was a bit tired-er than normal, but I mean, I was tearing around. My daughter was coming back for 3 months and they've got a house in the next village, I was over there doing things in the house, getting it ready.'* (F, 60–69 years, 2–4 weeks, cancer)

### Theme 2: responses to initial symptom appraisal

Participants reported a number of responses to initial symptom appraisal.

| Table 2 | Illustrative quotations relating to symptom attribution, a subtheme of theme 1— initial appraisal of symptoms |
| --- | --- |
| | **Initial appraisal of symptoms** |
| Diet and lifestyle | *'It comes and goes, it must be something that I eat and it just doesn't agree with me'* (F, 60–69 years, 13–26 weeks, non-cancer) <br> *'I think it's the picking up, it's the bending that does it a lot with my job as well, 'cos I lift heavy (clothing) all day, and I'm putting them over people and then bending and sitting down'* (F, 40–59 years, regular attender to GP, cancer) |
| Existing gastrointestinal conditions | *'I've always suffered with what's it called, you know, reflux, over the years, sometimes I could take some indigestion remedy and that Gaviscon thing and it'd calm down'* (F, 40–49 years, <2 weeks, non-cancer) |
| Other medical conditions, side effects of medication | *'At first I started putting it down to the diabetic pills, I thought 'Well it's probably because I've changed those', you know, and this Pioglitazone, I kept thinking 'Well it's that'* (F 60–69 years, 2–4 weeks, cancer) <br> *'I've had this sort of tiredness and a little bit of discomfort on and off for a time, I thought it was possibly diabetes…it kept on… and we thought was perhaps as a result of some of the medication I was taking for other things.'* (M, 70–79 years, 13–26 weeks, cancer) |

**Table 3**  Illustrative quotations relating to symptom progression, a subtheme of theme 3—further appraisal of symptoms

| | Further appraisal of symptoms |
|---|---|
| Pain | 'Well, it varied, you know, sometimes I'd have the pain all day and that's very wearing and painful…It's just under the rib cage and it's slightly to the right hand side, just slightly there, always in the same place.' (F, 70–79 years, <2 weeks, non-cancer)<br><br>'…I just had a nagging, more like a dull ache, underneath my bust and I thought it was indigestion and it didn't seem to go away, it didn't make any difference whether I ate or not, it was there quite a lot of the time' (F, 70–79 years, 2–4 weeks, cancer)<br><br>'After Christmas that then became all day, every day. I was going to bed with a stomach ache and I was waking up with a stomach ache…It was there all the while and you know, it didn't make sense that I'd have a problem all the while so I thought I was imagining it' (M, 60–69 years, 2–4 weeks, cancer) |
| Progressive changes in appetite and enjoyment of food | 'I was actually going off the actual taste of food, I wasn't enjoying food anymore, things that I would normally eat I wasn't enjoying' (M, 60–69 years, 5–8 weeks, cancer)<br><br>'Well it became quite traumatic at one point, when I just (did) not want an evening meal at all. I always eat a reasonable breakfast, but for the evening meal I had no appetite' (M, 80+ years, <2 weeks, cancer)<br><br>…and I was eating less and smaller meals and being really careful what I ate, and I thought 'it's good for you but you shouldn't have to be that precise when you eat' (F, 60–69 years, 26–52 weeks, non-cancer) |
| Weight loss | 'I was also suffering uncomfortable, you know, let's say stomach irritation, indigestion, wasn't happy eating, so I literally went along to say 'look, this isn't right, I don't mind losing weight but not like this'' (M 60–69 years, 5–8 weeks, cancer)<br><br>'I'd probably lost about four pounds which was concerning me but then I thought, you've reduced the amount that you're eating…' (F 50–59 years, 2–4 weeks, non-cancer) |

## Lifestyle adaptations

In the light of these attributions and explanations, some people began to make adaptations to their diet or daily living activities to alleviate their symptoms; in some cases, these changes were quite significant:

*'If I had smaller meals, but slightly more frequently, I could actually control keeping food, you know, keeping comfortable'* (M, 60–69 years, 5–8 weeks, cancer)

*'I think you learn to live with it and you just adapt to it and start to make the changes so it doesn't happen… I think what had happened with me is I couldn't control it any longer, no matter if I stopped eating things, didn't eat out, it was happening anyway'* (F, 60–69 years, 26–52 weeks, non-cancer)

## Role of others in appraisal and adaptations

Some people discussed their early symptoms with close relatives, particularly partners or children. This could prompt discussions around potential explanations and suggestions for improving symptoms, such as changes in content or timing of meals; at times it could also lead to delays in help-seeking:

*'When I mentioned to my wife the colour of my stools, which was the first symptom, she said I wasn't getting enough carbohydrates or protein or something…something wasn't right in my diet. So I changed my diet, I was eating fruit and vegetables and things like that, a lot more, and it still stayed the same.'* (M, 50–59 years, 13–26 weeks, cancer)

## Theme 3: further appraisal of symptoms

Changes in patterns of symptoms often led participants to further appraise their symptoms, and to question the reasons for their occurrence and their original explanations.

## Symptom progression (table 3)

Symptoms became of greater concern when they became more severe or frequent, or when further subtle symptoms developed. In some cases, it appeared that there was progression from subtle and intermittent symptoms of a manageable and tolerable nature, to symptoms which were more alarming such as pain. Across the whole sample, people described pain in a number of different parts of the body including their shoulder, chest, upper abdomen and back.

Another man described the chronology of his symptoms; having started with mild indigestion symptoms, additional, more alarming, symptoms developed:

*'gradually… the stools over a period of time, but the urine it seemed to happen more or less in a week, from normal colour to really deep colour'* (M, 50–59 years, 13–26 weeks, cancer)

Many participants reported subtle yet progressive changes to their appetite and enjoyment of food. Weight loss as a result of changes in appetite was also reported by a few participants.

## Symptom monitoring and self-management

The subtlety and slow progression of symptoms prompted some participants to monitor their symptoms, and some to consider alternative self-management strategies. Many used over-the-counter remedies to attempt to alleviate their symptoms of indigestion or abdominal pain:

*'It was only when I started feeling a bit of strangeness that I decided to try and keep a pattern of them'* (F, 60–69 years, 2–4 weeks, cancer)

### Impact on life

Most people described how their symptoms gradually or rapidly impacted on their quality of life and daily routine, including their social activities and hobbies, as well as their ability to sleep. Some felt that they were less able to complete tasks at home; for others, work became more challenging:

*'I decided I'd got this burning pain and I noticed that I was beginning to get a little bit short of breath, and also I couldn't do the same things I used to, like I found digging was becoming impossible'* (M, 70–79 years, 13–26 weeks, non-cancer)

### Previous family and personal history of cancer

A few people expressed a concern that they might develop cancer because they had lost a close relative to the disease; one had also previously suffered from cancer. These people were concerned about their personal risk of cancer and reported that they often considered whether their symptoms matched those which they expected from having cancer. Conversely one woman, subsequently diagnosed with a benign condition, had not considered cancer as a possibility because her symptoms did not match her expectations of cancer:

*'I knew I'm generally healthy, I mean I've been going swimming and to the gym…it wasn't seriously debilitating…I expect when you have cancer that you have more symptoms'* (F, 60–69 years, 26–52 weeks, non-cancer)

### Theme 4: deciding to seek help (table 4)

The further appraisal of symptoms eventually led most participants to decide to seek help from primary care, although several had emergency visits to A&E.

### Triggers

Triggers to seek help usually involved an increased frequency or intensity of symptoms, although sometimes it was precipitated by an alarming event, usually pain. Pain was often the 'tipping point' to seek help against a background of other more subtle symptoms. For those without pain, triggers were often complex, and could involve interpreting new or changing symptoms alongside symptoms from other existing conditions and, in some instances, symptoms from other acute illnesses. Healthcare factors, such as availability of a GP appointment, were seldom mentioned.

### Legitimising help-seeking

People were often encouraged to seek help by their spouse, family members and friends; they were particularly influenced by comments about changes in their general well-being, or recall of specific symptom episodes. A few participants reported considering whether their symptoms had become a 'legitimate' reason to seek healthcare when self-management strategies became ineffective; they tended to refer to the length of time symptoms had been experienced, and their reduced ability to cope.

## DISCUSSION

This is the first study to investigate symptom appraisal and help-seeking in patients newly referred to hospital with symptoms suggestive of pancreatic cancer. Due to their subtle and intermittent pattern, many people initially just monitored their symptoms. Symptoms were interpreted based on previous experiences with common conditions such as gastroenteritis, or impact of dietary change or existing gastrointestinal conditions. Many made changes to their eating patterns including type of food, and frequency and quantity of meals, or managed their symptoms with over-the-counter medication. Consequently, they tended not to go to their GP at that point. Changes to symptom frequency, duration or severity, their impact on daily life, or the appearance of additional symptoms, particularly pain, led many people to further appraise their symptoms and visit their GP. Family and friends often made important contributions to encouraging help-seeking by reflecting on changes in symptoms, although at times social contacts could reinforce self-management

| Table 4 | Illustrative quotations relating to theme 4: deciding to seek help |
|---|---|
| | **Illustrative quotations** |
| Pain on a background of more subtle symptoms | *'Sometimes I could take some indigestion remedy and… it'd calm down, but this were a different kind of pain, this were, oh it's excruciating pain…but that were the first time that… I thought there's something not right here and I need to get some help.'* (F, 40–49 years, <2 weeks, non-cancer) |
| Help-seeking encouraged by others | *'The kids had been saying 'mum you're always ill' and I thought 'oh yeah, I am'. Because you, I think you sort of drift, you get used to things and you don't see any problem with it, and when other people start to say, you know 'hang on' and then I thought 'oh yeah''* (F, 60–69 years, 26–52 weeks, non-cancer)<br>*'I'd fall asleep on my desk, you know, this isn't me, and other people said 'oh you're losing weight'… people actually were saying 'you don't look well'. It's another lever to go and see, to get it sorted'* (M, 60–69 years, 5–8 weeks, cancer) |
| Legitimate reason | *'I think my basic problem was pain… from arthritis that I'd had for some time, but it was proving impossible to control it by the methods I was using'* (M 80-89, 2-4 weeks, cancer) |

strategies. These patterns of symptom appraisal and help-seeking were seen regardless of final diagnosis.

## Strengths and limitations

The major strength of this study is that we prospectively recruited patients with symptoms suggestive of pancreatic cancer, and interviewed them during their pathway to diagnosis and treatment. This allowed detailed discussions about factors affecting symptom appraisal and help-seeking while they were still fresh events, and greatly reduced the possibility of post hoc rationalisation and recall bias. We have used similar approaches in interview studies with people recently diagnosed with melanoma,[22] and people with symptoms suggestive of lung[10] and colorectal[12] cancers. Recruiting patients during their diagnostic pathway enabled comparisons about symptom appraisal and help-seeking between people subsequently diagnosed with cancer and people with non-cancer conditions; importantly, we were not able to find differences between them. Our recruitment process meant that some patients were subsequently diagnosed with a cancer other than pancreatic cancer. We do not consider this a weakness of the study; rather, it serves to highlight the complexity and challenges of diagnosing pancreatic cancer, and how similarly a range of upper gastrointestinal cancers can present. While we aimed to recruit equal numbers of patients with pancreatic cancer and benign conditions, several patients with cancer withdrew before interview as they were too unwell.

Our approach to data collection and analysis was strengthened by the use of a strong theoretical framework: underpinned by recommendations from the Aarhus statement on improving design and reporting studies on early cancer diagnosis,[18] we applied the framework of the widely used Model of Pathways to Treatment,[16 17] and a specifically developed calendar-landmarking instrument which aided some participants' recall of dates of their symptoms and help-seeking.[14] Other strengths include recruitment from two English regions and a number of hospitals to ensure patients from a wide range of socioeconomic backgrounds, and who had entered hospital care via a variety of routes, enabling us to gather a broad range of experiences. Finally, the research team had a broad range of scientific and clinical expertise, and we sought inputs from our lay members at all stages of the research process, including analysis and interpretation of the data.

While we acknowledge that these experiences may not be representative of all people with symptoms suggestive of pancreatic cancer, the demographics of this cohort are similar to that of the main SYMPTOM Pancreas Study.[12] Although a major finding was that there were similar patterns in symptom appraisal and help-seeking between those diagnosed with cancer and those with other diagnoses, this may have been partly due to recruiting patients relatively late in their diagnostic pathway, when GPs had referred with a suspicion of pancreatic cancer. Ideally we would have recruited people with similar symptoms from earlier in their diagnostic pathway and from primary care, but there are significant challenges with the feasibility and costs of recruiting such a cohort containing sufficient cases of pancreatic cancer. The presence of relatives in some interviews may have led to potential bias in the participants' accounts and recollections, but this was minimised by only including participants' words in the analysis. Finally, we can only report the experiences shared with the researchers at the time of the interview: there may have been events which were not discussed if they were felt to be too emotional or private.

## Comparison with existing literature

Although these findings are in line with the broader literature investigating help-seeking for cancer symptoms,[10 12 22] very little has been published on symptom appraisal and help-seeking among patients diagnosed with pancreatic cancer. Evans *et al* recently reported the only other similar study, also set in England, and highlighted the intermittent and subtle nature of symptoms with their title: ''It can't be very important because it comes and goes'- patients' accounts of intermittent symptoms preceding a pancreatic cancer diagnosis'.[8] However, their interviews were conducted with people up to several years after a diagnosis of pancreatic cancer, and included data from interviews with relatives of people who had died. Our findings are strengthened by interviewing patients earlier in their illness pathway, thereby capturing subtle assessments of initial bodily changes and explanations, and minimising post hoc rationalisation and recall bias. We could not identify any subtle differences in the nature of symptoms or help-seeking between those who eventually were found to have pancreatic cancer, and those who were not. This is consistent with the quantitative findings of the SYMPTOM Pancreas Study,[9] where the evolution of additional symptoms such as fatigue, change in bowel habit, weight loss, decreased appetite and jaundice were associated with pancreatic cancer.

Other studies of pancreatic cancer symptoms have mainly used quantitative retrospective designs. For example, Stapley *et al* undertook a case-control study using routinely recorded UK primary care data which demonstrated how poorly predictive most symptoms are of pancreatic cancer in primary care populations. While the positive predictive value of jaundice was relatively high (21%), it was much lower for other symptoms including abdominal and back pain, nausea, vomiting, change in bowel habit, weight loss and malaise (<1%).[6] Keane *et al* more recently used another large UK primary care database to investigate the early symptom profiles of patients diagnosed with pancreatic ductal or biliary adenocarcinoma and found a similar wide range of early symptoms.[23] They also found that, in the year prior to diagnosis, these patients visited their GP on a median of 18 (IQR 11–27) occasions, which resonated with findings from another large English database study showing that 41% of patients diagnosed with pancreatic cancer had visited their GP at least three times before referral to specialist care.[7] In

contrast, more than half our sample reported only visiting their GP once before referral, suggesting either that our sample were recruited early in their symptom pathway, or perhaps that they had symptoms such as jaundice which are more likely to prompt early referral. Other quantitative studies set among patients newly diagnosed with pancreatic cancer have reported fatigue, dyspepsia and subtle changes in appetite such as early satiety in the preceding 6 months, confirming our finding that there may be a window of opportunity for timely investigations when patients report combinations of non-specific or subtle symptoms.[4 24 25]

### Implications for research and practice

Our findings add to current knowledge about the subtle and non-specific nature of early symptoms of pancreatic cancer, and enhance our understanding of their often intermittent and insidious nature. They highlight the importance of ongoing, and often iterative, appraisal of symptoms as described in the Model of Pathways to Treatment.[16 17] The findings have characterised pain as a 'tipping point' for presentation to healthcare, and underpinned the importance of family and friends in symptom monitoring, recognising evolving and new symptoms and endorsement of help-seeking.[26] The challenge for symptom awareness campaigns is to promote earlier presentation with these more subtle and intermittent symptoms which lead to changes in eating patterns including type of food, and frequency and quantity of meals, and weight loss. The 'Know 4 Sure' campaign was one of the first to raise awareness of non-site-specific symptoms including weight loss, pain, lump and unexplained bleeding. It was associated with increase in referrals for a range of cancers including upper gastrointestinal (GI) cancer.[27] Whether public health campaigns can effectively promote messages about combinations of non-specific symptoms is unknown, but may be important in pancreatic cancer.

The newly revised National Institute for Health and Care Excellence guidance on Suspected Cancer: Recognition and Referral[28] supports a lower threshold for investigating and referring people with single and multiple symptoms suggestive of pancreatic cancer, particularly those in higher-risk groups such as patients with newly diagnosed diabetes. The findings from this study could inform GP education approaches to promote timely investigation and referral for people with these non-specific and intermittent symptoms. These numerous subtle and non-specific symptoms, when combined with abnormal blood tests and the use of clinical decision support tools, might alert GPs to a possible pancreatic cancer diagnosis earlier. Safety-netting, defined as is a diagnostic strategy or consultation technique to help manage diagnostic uncertainty and ensure that patients are followed up in a timely and appropriate manner, is also important due to the evolving nature of symptoms; appropriate strategies would include symptom monitoring and early review.

In conclusion, this study has described and contextualised the appraisal and help-seeking behaviours of patients with symptoms that could represent pancreatic cancer. Greater awareness of combinations of the subtle and intermittent symptoms, and their evolving nature, is needed to prompt timelier help-seeking and investigation among people with symptoms of pancreatic cancer.

**Acknowledgements** The authors thank the patients who so kindly and freely gave their time and experiences to this study. The authors thank Helen Morris the SYMPTOM Study manager, the other study patient and public representatives Sue Ballard and Victor Boulter, and the staff at the participating hospitals (Bedford Hospital NHS Trust, Cambridge University Hospitals NHS Foundation Trust; Hinchingbooke Healthcare NHS Trust, The Queen Elizabeth Hospital Kings Lynn NHS Trust, Peterborough and Stamford NHS Foundation Trust, University Hospital of North Durham; Darlington Memorial Hospital: University Hospital of North Tees, West Suffolk NHS Foundation Trust) who facilitated participant recruitment and supported the study. The authors also thank the NIHR funded Discovery Programme Steering Committee comprising: Roger Jones (chair); Alison Clutterbuck; Ardiana Gjini; Joanne Hartland; Maire Justice; Jenny Knowles; Richard Neal; Peter Rose, for their contribution.

**Contributors** FMW, JDE, GPR and WH were responsible for study design, study conduct and management. KM, LB and NH collected the data. With support from JB, JDE, JL, MJ and FMW, KM analysed and interpreted the data. KM led the manuscript preparation. All authors contributed to the writing and completion of the paper and have approved the final the manuscript.

**Funding** This paper presents independent research funded by the National Institute for Health Research Programme Grants for Applied Research programme (RP-PG-0608-10045) and Pancreatic Cancer Action (registered charity 1137689). The views expressed are those of the authors and not necessarily those of the NHS, the NIHR or the Department of Health. The funders of this study had no role in study design, data collection, analysis, and interpretation, or the writing of this research paper. FMW is supported by an NIHR Clinician Scientist award.

**Competing interests** None declared.

**Patient consent** Obtained.

**Ethics approval** The Cambridgeshire 3 Research Ethics Committee (10/H0306/50).

**Provenance and peer review** Not commissioned; externally peer reviewed.

**Data sharing statement** No additional data are available.

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
