## [Reviewer comments · BMJ Open]

ARTICLE DETAILS

TITLE (PROVISIONAL)	Understanding symptom appraisal and help-seeking in people with symptoms suggestive of pancreatic cancer: a qualitative study.
AUTHORS	Mills, Katie; Birt, Linda; Emery, Jon; Hall, Nicola; Banks, Jonathan; Johnson, Margaret; Lancaster, John; Hamilton, Willie; Rubin, Greg; Walter, Fiona

VERSION 1 - REVIEW

REVIEWER	Jennifer Fish Flinders University, Australia
REVIEW RETURNED	08-Mar-2017

GENERAL COMMENTS	The manuscript presents an interesting exploration of symptom appraisal and help-seeking behaviour among people newly referred to hospital with symptoms suggestive of pancreatic cancer. This paper is a useful addition to the literature. I have some comments about clarity though. Abstract: Written expression could be improved in parts of the abstract and there are some errors. For example, Line 17, "in depth- interviews" should be "in-depth interviews" with a full stop following. Sentences starting with numbers should be written out in full rather than using numerical digits. - Design: The type of analysis is partially reported under the 'Participants and setting' heading, but it would be more usefully described in full under the 'Design' heading.- Participants and Setting: It would be useful to report the selection criteria and the number of hospital sites in this section.- Results: It would be useful to report how many participants were diagnosed with cancer and how many with other conditions. This information is important considering the results presented. It might also be better reported under the 'Participants and setting' heading. Considering the data were analysed thematically, it would be useful to report the themes in the results section. Strengths and limitations: - In addition to the strengths of the study, it is important to thoughtfully consider the limitations of the study here.- The first bullet point is very lengthy; reporting results here might not be necessary.
---

Introduction:

Written expression, clarity, and referencing could be improved in parts of the introduction.

- Line 25: This sentence requires a reference.
- Line 34: The studies mentioned in this sentence require references.
- Paragraph two is somewhat difficult to follow; structure and clarity of argument could be improved.
- Line 54: The authors mention a few studies, but only reference one. Either clarity or referencing requires improvement.
- As one of the key results is a comparison across people who were diagnosed with cancer and those with other conditions, it might be useful to introduce why this examination is important in the introduction.

Methods:

- The qualitative approach (e.g., grounded theory, qualitative description) and rationale is not stated in the method section. A brief outline of the approach would be beneficial.
- Recruitment: The description of recruitment was relatively clear, however, the rationale could be briefly addressed (e.g., why participants aged 40 years or over). Also, how were participants invited to participate in an interview?
- Data collection: The authors report that relatives or spouses were included in several interviews, however, the consent process for spouses is not clear. Did spouses provide informed written consent?
- Line 30: A page number for this quote would be beneficial.
- Analytic process: It is unclear how each of the three authors (KM, LB, NH) worked together to apply the Framework analysis, and how the patient representatives contributed to all stages of the analysis. Further detail would be beneficial. Was charting completed as part of the analytic process? Also, reflexivity was only minimally evident in the discussion. The influence of researcher characteristics on the study and analysis might have been more thoughtfully addressed.

Results:

- Written expression could be improved in parts of the results. Line 6: Again, sentences starting with numbers should be written out in full rather than using numerical digits, or be re-worded. Line 9: Write numbers one to nine in full rather than using numerical digits. Line 10: There is a full stop missing at the end of the sentence. Please proof read the paper carefully.
- Sample: Reporting of the timeframe when participants were interviewed seems to be incomplete. The authors report that six participants were interviewed prior to diagnosis and 12 within four weeks of diagnosis. When were the remaining eight participants interviewed? The authors refer to Table 1 for more information, but there is no relevant information in the table?
- Theme 1: It would be useful to list all initial symptoms experienced by participants early in Theme 1.
- Page 9, Line 7: How many participants were affected by other conditions?
- In the 'Methods' section the authors report that they examined the data according to participant characteristics, including age, gender, co-morbidity, geographical region, and diagnosis. However, there was no mention of age, gender, or region in the results. Were there

	any differences according to these characteristics? For example, were there any differences according to gender in the 'role of others in appraisal and adaptations'? Even if there were no differences according to these characteristics, it would be useful to report this. - For illustrative quotes it might be useful to differentiate between participants affected by pancreatic cancer and other cancers. Discussion: - Since very little has been published on symptom appraisal and help-seeking behaviour among people diagnosed with pancreatic cancer, the authors could briefly compare with the wider literature investigating help-seeking for cancer symptoms. Are the study findings in line with the broader cancer literature? - Recall issues might be more thoughtfully addressed considering the wide ranging TTP in the study. References: - Reference 21: The year is missing for this reference.
--	--

REVIEWER	Dr Afrodita Marcu University of Surrey, UK
REVIEW RETURNED	10-Mar-2017

GENERAL COMMENTS	Major comments: 1. The Introduction is underdeveloped and needs to include more literature review on pancreatic or other types of gastro-intestinal cancers where early diagnosis is difficult. Given that in the present study not all the participants were diagnosed with pancreatic cancer, the Introduction should be broader in scope and cover the other types of cancer (e.g. gallbladder) where the symptoms are similar to pancreatic, or equally difficult to detect early. The reference in the Methods to "similar studies having been conducted for people with symptoms suggestive of lung and colorectal cancer [10-12]" (p.5) belongs better in the Introduction. 2. The study rationale could be better formulated, as the timing of the qualitative exploration cannot by itself improve symptom appraisal or diagnostic delays in pancreatic cancer. E.g. why there is merit in exploring symptom appraisal and help-seeking for pancreatic cancer prior to being given a cancer diagnosis. 3. It should be mentioned in the Abstract that only half of the participants were subsequently diagnosed with cancer, and of these only 9 had pancreatic cancer. There needs to be more clarity on the recruitment process and inclusion criteria, e.g. why were the participants eligible only if they were aged 40 or older? (p.5) Secondly, if the participants were recruited prior to receiving a diagnosis of cancer / other diagnosis, then their diagnosis characteristics could not have been known at the point of recruitment, rather, their "symptom" characteristics. Also, it needs to be mentioned more explicitly in the Methods whether the participants knew, at the time of the interview, whether they had a diagnosis of cancer or of another condition. Seven participants were unaware of their diagnosis at interview (p.7), yet 6 were interviewed prior to diagnosis (p.8) – the numbers don't seem to tally. Were there any differences in patients' accounts
---

of symptom attribution and help-seeking between the participants who knew their diagnosis and those who did not at the time of the interview? Table 1 (or an additional table) should clearly indicate how many patients had a cancer vs. other diagnosis, and of these how many knew their diagnosis at the point of interview. In the analysis it is mentioned that some patients had had a history of cancer – if so, how many and what type of cancer? This characteristic needs to be made explicit in the Results or in Table 1. How did a previous experience of cancer influence symptom appraisal? Did the participants consider cancer as a cause for their symptoms?

4. The heading “theme” and the categories included under “theme” in the tables with quotes are confusing: they can be mistaken for the analytical themes of the framework analysis, when in fact they are “codes”. The summary of the themes and sub-themes could be included in a table. Some of the quotes in Table 2, e.g. for “normalisation” are inappropriate and do not support the concept of “normalisation” as described in the paragraph on p.9 – see recent examples on normalization in the literature, e.g. Whitaker et al. 2015, Br J Gen Pract, DOI: 10.3399/bjgp15X683533; Low et al. 2015, BMJ Open, 5:e008082, doi:10.1136/bmjopen-2015-008082; Marcu et al. 2017, Br J Health Psych, DOI: 10.1111/bjhp.12215. The overall analysis needs to be refined; for example, the sub-theme “Lifestyle adaptations” does not seem to fit in with the theme “Initial appraisal of symptoms”. The second theme, “Further appraisal of symptoms”, describes the symptoms but says little about how the patients changed their symptom appraisal or coping strategies. Theme 3, “Deciding to seek help”, is under-developed compared to the other two themes, yet it is an important one because it sheds light on what eventually leads people to seek help for these symptoms.

5. In the Discussion there needs to be more reflection on subsequent diagnosis of the participants with cancer vs. non-cancer, otherwise the paper reads as if all the participants had symptoms indicative of pancreatic cancer. The comparison of the present findings with previous literature should be included in the Introduction instead of the Discussion – indeed, the scarcity of qualitative research on pancreatic cancer would help to set the scene. The Discussion should only compare the findings, not the methodological approaches, of similar studies.

Minor comments:

1. Symptoms “suggestive” or “indicative”, rather than “suspicious”, of pancreatic cancer (in the Abstract and throughout the paper)

2. The Aarhus statement does not “recommend” the Model of Pathways to Treatment – it simply recommends a different conceptualisation of patient delay (i.e. “patient interval”). The MPT builds on the Aarhus statement.

3. The interview topic guide should have been included as an appendix, or in a table. The authors should mention whether the interview guide was adapted for those participants who did not know their diagnosis at the time of the interview.

4. Some reflection needed on fidelity of the interviewing, as three different researchers conducted the interviews. Were they all

	experienced qualitative researchers? 5. Did the study patient representatives have a direct experience of pancreatic cancer? And did their contribution to the analysis draw on that? e.g. did the themes reflect their personal experience of pancreatic cancer?
--	---

VERSION 1 – AUTHOR RESPONSE

Reviewer #1:

The manuscript presents an interesting exploration of symptom appraisal and help-seeking behaviour among people newly referred to hospital with symptoms suggestive of pancreatic cancer. This paper is a useful addition to the literature. I have some comments about clarity though.

We appreciate the reviewer's comments and have made amendments to the manuscript as suggested.

Abstract:

Written expression could be improved in parts of the abstract and there are some errors. For example, Line 17, "in depth- interviews" should be "in-depth interviews" with a full stop following. Sentences starting with numbers should be written out in full rather than using numerical digits.

- Design: The type of analysis is partially reported under the 'Participants and setting' heading, but it would be more usefully described in full under the 'Design' heading.

We thank the reviewer for these comments and have made these changes in the abstract.

- Participants and Setting: It would be useful to report the selection criteria and the number of hospital sites in this section.
- Results: It would be useful to report how many participants were diagnosed with cancer and how many with other conditions. This information is important considering the results presented. It might also be better reported under the 'Participants and setting' heading. Considering the data were analysed thematically, it would be useful to report the themes in the results section.

We have added the number of hospital sites and inclusion criteria, so the initial sentence in the participants and setting subheading of the abstract now reads:

Patients aged ≥ 40 years were recruited from nine hospitals after being referred to hospital with symptoms suspicious of pancreatic cancer; all were participants in a cohort study.

We have also made spelling corrections. However, the limited words available for the abstract do not allow us to add any more detail.

Strengths and limitations:

- In addition to the strengths of the study, it is important to thoughtfully consider the limitations of the study here.
- The first bullet point is very lengthy; reporting results here might not be necessary.

We don't believe that we can shorten the first bullet point but, as suggested, have added one further bullet point about limitations:

Some of the most seriously ill patients were unable to be interviewed and their experiences may have differed from those in our sample.

Introduction:

Written expression, clarity, and referencing could be improved in parts of the introduction.

- Line 25: This sentence requires a reference.
- Line 34: The studies mentioned in this sentence require references.
- Paragraph two is somewhat difficult to follow; structure and clarity of argument could be improved.
- Line 54: The authors mention a few studies, but only reference one. Either clarity or referencing requires improvement.

All these suggested changes have been made.

- As one of the key results is a comparison across people who were diagnosed with cancer and those with other conditions, it might be useful to introduce why this examination is important in the introduction.

We have extended the third paragraph to reflect the comments of both Reviewers:

Furthermore, many symptoms of pancreatic cancer are also symptoms of other gastrointestinal or intra-abdominal cancers such as gallbladder, colon and ovarian cancer, which may be equally difficult to diagnose early [7].

Methods:

- The qualitative approach (e.g., grounded theory, qualitative description) and rationale is not stated in the method section. A brief outline of the approach would be beneficial.

Thank you- we have added a brief outline of the analytical approaches:

The analytic process was iterative, starting after the first few interviews. Three researchers with social sciences expertise (KM, LB, NH) worked together, initially using thematic analysis techniques [19] to look for unexpected inductive codes and categories, and then adopting a framework approach [20] guided by the stages of the Model of Pathways to Treatment [16.17]. These approaches were combined to ensure a rigorous and systematic progression through the five analytic steps:

- Recruitment: The description of recruitment was relatively clear, however, the rationale could be briefly addressed (e.g., why participants aged 40 years or over). Also, how were participants invited to participate in an interview?

The rationale has already been described in other papers arising from the same study and referenced at the beginning of the Methods section (9-12). We have added further explanation about the invitation:

They were initially sent an invitation letter and SYMPTOM study questionnaire by post; people who completed the questionnaire were able to indicate whether they would be willing to take part in an interview.

- Data collection: The authors report that relatives or spouses were included in several interviews, however, the consent process for spouses is not clear. Did spouses provide informed written consent?

This point has been clarified:

A relative, usually a spouse, was also present at several interviews at the participant's request: they assisted in recall of events and confirmed participants' accounts, but only the participant's words were included as data in analyses.

- Line 30: A page number for this quote would be beneficial.

This has been included.

- Analytic process: It is unclear how each of the three authors (KM, LB, NH) worked together to apply the Framework analysis, and how the patient representatives contributed to all stages of the analysis. Further detail would be beneficial. Was charting completed as part of the analytic process? Also, reflexivity was only minimally evident in the discussion. The influence of researcher characteristics on the study and analysis might have been more thoughtfully addressed.

We have added further detail to this section in response to the Reviewer's helpful comments as above and here:

Finally, the researchers examined patient data by characteristics, including age, gender, co-morbidity, geographical region and diagnosis, to seek patterns and non-confirming cases. The study patient representatives (MJ, JL) contributed to all stages of analysis by reading transcripts and regular meetings with the research team, and the other authors (all with clinical or social science expertise) contributed to the interpretation of the data.

Results:

- Written expression could be improved in parts of the results. Line 6: Again, sentences starting with numbers should be written out in full rather than using numerical digits, or be re-worded. Line 9: Write numbers one to nine in full rather than using numerical digits. Line 10: There is a full stop missing at the end of the sentence. Please proof read the paper carefully.

We have made changes to the manuscript as suggested.

- Sample: Reporting of the timeframe when participants were interviewed seems to be incomplete. The authors report that six participants were interviewed prior to diagnosis and 12 within four weeks of diagnosis. When were the remaining eight participants interviewed? The authors refer to Table 1 for more information, but there is no relevant information in the table?

Thank you for noting these oversights: we have now amended as follows:

All participants were interviewed soon after their referral to specialist care (range: 1 day to 18 weeks), with a quarter (6, 23%) prior to diagnosis, half (12, 46%) within 4 weeks of diagnosis, and the remainder within 18 weeks (8, 31%).

- Theme 1: It would be useful to list all initial symptoms experienced by participants early in Theme 1.

While we appreciate this comment we feel that initial subtle or non-specific bodily changes such as feeling tired or 'different' are well covered in Theme 1.

- Page 9, Line 7: How many participants were affected by other conditions?

This line has been amended to read:

Nine participants with other conditions....

- In the 'Methods' section the authors report that they examined the data according to participant characteristics, including age, gender, co-morbidity, geographical region, and diagnosis. However, there was no mention of age, gender, or region in the results. Were there any differences according to these characteristics? For example, were there any differences according to gender in the 'role of others in appraisal and adaptations'? Even if there were no differences according to these characteristics, it would be useful to report this.

This is a good point. We have amended the Qualitative themes section to include:

We were unable to find any clear differences regarding symptom appraisal or help-seeking between the accounts of people diagnosed with cancer and those with other conditions. We therefore report the findings together. We were also unable to find any clear differences according to patient characteristics including gender, age and region.

- For illustrative quotes it might be useful to differentiate between participants affected by pancreatic cancer and other cancers.

We did not find any differences between participants affected by pancreatic cancer and other cancers, nor between those affected by cancer and other conditions. We therefore do not consider this would be helpful; furthermore, it may make participants more identifiable.

Discussion:

- Since very little has been published on symptom appraisal and help-seeking behaviour among people diagnosed with pancreatic cancer, the authors could briefly compare with the wider literature investigating help-seeking for cancer symptoms. Are the study findings in line with the broader cancer literature?

This is a helpful point too. We have added a sentence at the beginning of the section on comparisons with existing literature:

Although these findings are in line with the broader literature investigating help-seeking for cancer symptoms [10,12, 22], very little has been published on symptom appraisal and help-seeking among patients diagnosed with pancreatic cancer.

- Recall issues might be more thoughtfully addressed considering the wide ranging TTP in the study.

Thank you- we believe that this has already been addressed in the first section of Strengths and limitations:

The major strength of this study is that we prospectively recruited patients with symptoms suggestive of pancreatic cancer, and interviewed them during their pathway to diagnosis and treatment.

This allowed detailed discussions about factors affecting symptom appraisal and help-seeking while they were still fresh events, and greatly reduced the possibility of post-hoc rationalisation and recall bias.

Reviewer #2:

I thank the Editor for inviting me to review this interesting manuscript which addresses a very important topic in public health. This manuscript is concerned with symptom appraisal and help-seeking strategies among patients experiencing symptoms indicative of pancreatic cancer. I found the manuscript informative and stimulating. The strengths of the manuscript lie in its recruitment strategy, clear theoretical framework, and originality of the topic. While the manuscript is generally very well-written, the analysis and the presentation of the results could be improved. Please see my suggestions below in this sense.

We are very grateful to the reviewer for these supportive comments.

Major comments:

1. The Introduction is underdeveloped and needs to include more literature review on pancreatic or other types of gastro-intestinal cancers where early diagnosis is difficult. Given that in the present study not all the participants were diagnosed with pancreatic cancer, the Introduction should be broader in scope and cover the other types of cancer (e.g. gallbladder) where the symptoms are similar to pancreatic, or equally difficult to detect early. The reference in the Methods to “similar studies having been conducted for people with symptoms suggestive of lung and colorectal cancer [10-12]” (p.5) belongs better in the Introduction.

In response to similar helpful comments from Reviewer 1 and Reviewer 2 we have extended the third paragraph of the Introduction- see above. However, we don't agree that “similar studies having been conducted for people with symptoms suggestive of lung and colorectal cancer [10-12]” (p.5) should be moved from the Methods to the Introduction as it refers to similar methods of data collection.

2. The study rationale could be better formulated, as the timing of the qualitative exploration cannot by itself improve symptom appraisal or diagnostic delays in pancreatic cancer. E.g. why there is merit in exploring symptom appraisal and help-seeking for pancreatic cancer prior to being given a cancer diagnosis.

This has been clarified as:

The aim of this study was therefore to gain understanding of barriers and facilitators to symptom appraisal and help-seeking decisions among patients with symptoms suggestive of pancreatic cancer much earlier in their diagnostic pathway, to contribute to the development of interventions to promote timely diagnosis. We recruited patients who were newly referred to hospital, thus providing the opportunity to investigate the complexities of patients' symptom appraisal and decision making, and explore pathways from their first symptom to first presentation in primary care, emergency presentations, and referrals to specialist services.

3. It should be mentioned in the Abstract that only half of the participants were subsequently diagnosed with cancer, and of these only 9 had pancreatic cancer. There needs to be more clarity on the recruitment process and inclusion criteria, e.g. why were the participants eligible only if they were aged 40 or older? (p.5) Secondly, if the participants were recruited prior to receiving a diagnosis of cancer / other diagnosis, then their diagnosis characteristics could not have been known at the point of recruitment, rather, their “symptom” characteristics.

Also, it needs to be mentioned more explicitly in the Methods whether the participants knew, at the time of the interview, whether they had a diagnosis of cancer or of another condition.

These points have all been addressed in response to Reviewer 1's comments- see above.

Seven participants were unaware of their diagnosis at interview (p.7), yet 6 were interviewed prior to diagnosis (p.8) – the numbers don't seem to tally.

Thank you for pointing out this inconsistency- it has been corrected on p7 to:

Six were unaware of their diagnosis at interview.

Were there any differences in patients' accounts of symptom attribution and help-seeking between the participants who knew their diagnosis and those who did not at the time of the interview?

We were unable to find any evidence of such differences; therefore, it was not referred to in the manuscript. A sentence has been added to the Results section as follows:

We were unable to find any clear differences regarding symptom appraisal or help-seeking between the accounts of people diagnosed with cancer and those with other conditions. We therefore report the findings together. We were also unable to find any clear differences according to whether or not participants knew their diagnosis at the time of the interview, or their characteristics including gender, age and region.

Table 1 (or an additional table) should clearly indicate how many patients had a cancer vs. other diagnosis, and of these how many knew their diagnosis at the point of interview.

Table 1 already describes the cancer and non-cancer diagnoses of all the participants. In view of our response above, we do not think that further details need to be added to the Table.

In the analysis it is mentioned that some patients had had a history of cancer – if so, how many and what type of cancer? This characteristic needs to be made explicit in the Results or in Table 1. How did a previous experience of cancer influence symptom appraisal? Did the participants consider cancer as a cause for their symptoms?

In fact, we mentioned that a few people felt at risk of cancer due to a previous family history of cancer. Only one participant had a personal history of cancer and this has been clarified.

A few people expressed a concern that they might develop cancer because they had lost a close relative to the disease; one had also previously suffered from cancer.

We do not feel that this needs to be made more explicit in the Results or in Table 1 (and it may make the participant identifiable). We do believe that the presented data explains how the participants with affected relatives may have considered cancer as a cause for their symptoms.

These people were concerned about their personal risk of cancer and reported that they often considered whether their symptoms matched those which they expected from having cancer. Conversely one woman, subsequently diagnosed with a benign condition, had not considered cancer as a possibility because her symptoms did not match her expectations of cancer:

4. The heading “theme” and the categories included under “theme” in the tables with quotes are confusing: they can be mistaken for the analytical themes of the framework analysis, when in fact they are “codes”. The summary of the themes and sub-themes could be included in a table. Some of the quotes in Table 2, e.g. for “normalisation” are inappropriate and do not support the concept of “normalisation” as described in the paragraph on p.9 – see recent examples on normalization in the literature, e.g. Whitaker et al. 2015, Br J Gen Pract, DOI: 10.3399/bjgp15X683533; Low et al. 2015, BMJ Open, 5:e008082, doi:10.1136/bmjopen-2015-008082; Marcu et al. 2017, Br J Health Psych, DOI: 10.1111/bjhp.12215.

We agree with the Reviewer's thoughtful reflections and have therefore:

- (1) omitted the word Theme from each of Tables 2, 3 and 4, and
- (2) removed the quotations from Table 2 relating to 'normalisation'.

As the Reviewer suggests, we have made the point about 'normalisation' sufficiently in the paragraph on p.9.

The overall analysis needs to be refined; for example, the sub-theme "Lifestyle adaptations" does not seem to fit in with the theme "Initial appraisal of symptoms". The second theme, "Further appraisal of symptoms", describes the symptoms but says little about how the patients changed their symptom appraisal or coping strategies. Theme 3, "Deciding to seek help", is under-developed compared to the other two themes, yet it is an important one because it sheds light on what eventually leads people to seek help for these symptoms.

Again, we are very grateful for the Reviewer's thoughtful reflections. We have refined the analysis into the following headings, and developed Theme 3 (now Theme 4) into two sub-themes:

1. Initial appraisal of symptoms
 - 1.1 Bodily changes
 - 1.2 Symptom attributions (Table 2)
2. Responses to initial symptom appraisal
 - 2.1 Lifestyle adaptations
 - 2.2 Role of others in appraisal and adaptations
3. Further appraisal of symptoms
 - 3.1 Symptom progression (Table 3)
 - 3.2 Symptom monitoring and self-management
 - 3.3 Impact on life
 - 3.4 Previous family and personal history of cancer
4. Deciding to seek help (Table 4)
 - 4.1 Triggers
 - 4.2 Legitimising help-seeking

5. In the Discussion there needs to be more reflection on subsequent diagnosis of the participants with cancer vs. non-cancer, otherwise the paper reads as if all the participants had symptoms indicative of pancreatic cancer.

The participants did all have symptoms indicative of pancreatic cancer- this was the recruitment route. They subsequently went on to have cancer and non-cancer diagnoses as we would expect for all patients referred with these symptoms.

The comparison of the present findings with previous literature should be included in the Introduction instead of the Discussion – indeed, the scarcity of qualitative research on pancreatic cancer would help to set the scene. The Discussion should only compare the findings, not the methodological approaches, of similar studies.

Thank you- we believe that this is now addressed in the revised paper.

Minor comments:

1. Symptoms "suggestive" or "indicative", rather than "suspicious", of pancreatic cancer (in the Abstract and throughout the paper)

We have amended the paper as suggested.

2. The Aarhus statement does not “recommend” the Model of Pathways to Treatment – it simply recommends a different conceptualisation of patient delay (i.e. “patient interval”). The MPT builds on the Aarhus statement.

The MPT was developed prior to the Aarhus statement and the authors of the MPT were also authors of the Aarhus statement. We are not clear what the reviewer means here.

3. The interview topic guide should have been included as an appendix, or in a table. The authors should mention whether the interview guide was adapted for those participants who did not know their diagnosis at the time of the interview.

The interview topic guide has been added as an appendix.

4. Some reflection needed on fidelity of the interviewing, as three different researchers conducted the interviews. Were they all experienced qualitative researchers?

All researchers were experienced qualitative researchers.

5. Did the study patient representatives have a direct experience of pancreatic cancer? And did their contribution to the analysis draw on that? e.g. did the themes reflect their personal experience of pancreatic cancer?

The patient representatives have direct experience of cancer, one with a direct experience of pancreatic cancer. It would be reasonable to accept that their personal experience would be reflected in their interpretation of the data collected

VERSION 2 – REVIEW

REVIEWER	Dr Afrodita Marcu University of Surrey, UK
REVIEW RETURNED	16-May-2017

GENERAL COMMENTS	ABSTRACT 1. I agree with Reviewer#1 that not enough details are provided on the recruitment of participants, e.g. why 40 years old or older. 2. In the Results, it is not clear how many participants were subsequently diagnosed with pancreatic cancer, and how many with other diseases. This needs to be made explicit. If space is an issue, take out the description of time to first presentation. 3. The Results should present a summary of the themes, not a general description of the pathway to diagnosis. INTRODUCTION 1. I still believe that the paragraph “similar studies having been conducted for people with symptoms suggestive of lung and colorectal cancer [10-12]” (p.5) belongs better in the Introduction than in the Methods. The paragraph does not shed light on what kinds of methods were employed in the studies on lung or colorectal
--

cancer - this paragraph should be about the value of exploring patients' symptom appraisal and help-seeking and how this can contribute to earlier diagnosis of cancer.

METHODS

1. Re analytic process(p.7): It is incorrect to say that researchers can look for "unexpected inductive codes and categories". Firstly, you cannot deliberately look for something you don't expect; secondly, inductive codes and categories are not searched for, but constructed by the researcher during the analytic process. It is called "inductive coding", or data-driven coding. The authors would do well to refresh their knowledge of thematic analysis and indeed of the terminology associated with the method – I recommend reading Braun & Clarke's papers (2006; 2016). The use of framework analysis after an initial inductive thematic analysis suggests a deductive approach – this needs further clarification, e.g. were themes constructed to reflect symptom appraisal, help-seeking, and other stages of the MPT?

2. Recruitment: I agree with Reviewer#1 that the recruitment of the participants is not clear in the Methods section. The authors should explain why they recruited only patients aged 40 or older, even if it means restating what has been written in the previous studies. It is not enough to say that the rationale has already been described in other papers arising from the same study – the rationale and the recruitment strategy need to be made explicit in the present paper.

3. Data collection and the presence of a relative/spouse at the interview: the authors need to state clearly whether the participants' relatives/spouse provided consent or not (fine if not). The authors also need to reflect on how the relatives/spouses might have led to potential bias in the participants' accounts and recollection.

4. The patient representatives' role: Still not clear whether the patients' representatives contributed to the data analysis itself – this needs to be clarified. E.g. did they agree that the themes identified reflected their own personal experience of (pancreatic) cancer? In fact, did the patient representatives have direct experience of pancreatic cancer? Virtually no background is provided on them, therefore it is difficult to judge how they contributed to the study design or data analysis.

The authors state in their response letter that: "The patient representatives have direct experience of cancer, one with a direct experience of pancreatic cancer. It would be reasonable to accept that their personal experience would be reflected in their interpretation of the data collection". However, such detail needs to be included in the manuscript, not in the rebuttal letter. Furthermore, whilst it might be "reasonable to accept that their personal experience would be reflected in their interpretation of the data collection", this needs to be made explicit in the manuscript.

5. I could not see the interview topic guide attached as an appendix - and there is no reference in the Methods section about the topic guide being in an appendix. The authors need to include the topic guide and refer to it in the manuscript.

RESULTS

1. I disagree with the authors that Table 1 does not need further details or changes. It would be more informative if the column on the

	SYMPTOM Pancreas study were removed and instead the demographics and other details of the participants were presented according to cancer diagnosis vs. non-cancer diagnosis (even if there are no major differences between the two groups). 2. Overall, the new structure of the themes and sub-themes is much clearer than before. However, simply removing the word "Theme" from Tables 2, 3 and 4 does not solve the problem – what are those words and phrases meant to represent? Are they codes? Are they sub-themes? They need a heading to explain what they represent. Table 2 is not about "initial explanations" but rather about "symptom attributions". Perhaps Theme 2 (Lifestyle adaptations) should have a table of illustrative quotes, like the other themes. DISCUSSION 1. Overall, the Discussion is well written and includes good reflection on the strengths and limitations of the study. It would be good though to clarify, were possible, those "subsequently" diagnosed with pancreatic cancer vs. those "subsequently" diagnosed with other non-cancer conditions, as I presume that the final diagnosis occurred after the interviews were conducted. 2. A reference is needed to the wider SYMPTOM study (paragraph 2, p. 16). 3. It might be useful to define safety-netting (p.18) and what this would mean in the context of pancreatic cancer.
--	--

VERSION 2 – AUTHOR RESPONSE

Reviewer: 2 Dr Afrodita Marcu, University of Surrey, UK

The authors have addressed some of the changes suggested by the reviewers but there are some areas which still need clarification or change. I detail these below by manuscript section:

ABSTRACT

1. I agree with Reviewer#1 that not enough details are provided on the recruitment of participants, e.g. why 40 years old or older.
2. In the Results, it is not clear how many participants were subsequently diagnosed with pancreatic cancer, and how many with other diseases. This needs to be made explicit. If space is an issue, take out the description of time to first presentation.
3. The Results should present a summary of the themes, not a general description of the pathway to diagnosis.

These three points have all been addressed in the Abstract.

INTRODUCTION

1. I still believe that the paragraph "similar studies having been conducted for people with symptoms suggestive of lung and colorectal cancer [10-12]" (p.5) belongs better in the Introduction than in the Methods. The paragraph does not shed light on what kinds of methods were employed in the studies on lung or colorectal cancer - this paragraph should be about the value of exploring patients' symptom appraisal and help-seeking and how this can contribute to earlier diagnosis of cancer.

This sentence has been moved from the Methods to the Introduction, and the sentence refined to reflect the reviewer's suggestion.

METHODS

1. Re-analytic process (p.7): It is incorrect to say that researchers can look for “unexpected inductive codes and categories”. Firstly, you cannot deliberately look for something you don’t expect; secondly, inductive codes and categories are not searched for, but constructed by the researcher during the analytic process. It is called “inductive coding”, or data-driven coding. The authors would do well to refresh their knowledge of thematic analysis and indeed of the terminology associated with the method – I recommend reading Braun & Clarke’s papers (2006; 2016). The use of framework analysis after an initial inductive thematic analysis suggests a deductive approach – this needs further clarification, e.g. were themes constructed to reflect symptom appraisal, help-seeking, and other stages of the MPT?

The analytic process has been revised and clarified following the reviewer’s guidance.

2. Recruitment: I agree with Reviewer#1 that the recruitment of the participants is not clear in the Methods section. The authors should explain why they recruited only patients aged 40 or older, even if it means restating what has been written in the previous studies. It is not enough to say that the rationale has already been described in other papers arising from the same study – the rationale and the recruitment strategy need to be made explicit in the present paper.

A further sentence has been added to the Methods section (as in the Abstract) to clarify the participant recruitment.

3. Data collection and the presence of a relative/spouse at the interview: the authors need to state clearly whether the participants’ relatives/spouse provided consent or not (fine if not). The authors also need to reflect on how the relatives/spouses might have led to potential bias in the participants’ accounts and recollection.

Clarifications have been made to the Methods and the Discussion as suggested.

4. The patient representatives’ role: Still not clear whether the patients’ representatives contributed to the data analysis itself – this needs to be clarified. E.g. did they agree that the themes identified reflected their own personal experience of (pancreatic) cancer? In fact, did the patient representatives have direct experience of pancreatic cancer? Virtually no background is provided on them, therefore it is difficult to judge how they contributed to the study design or data analysis.

The authors state in their response letter that: “The patient representatives have direct experience of cancer, one with a direct experience of pancreatic cancer. It would be reasonable to accept that their personal experience would be reflected in their interpretation of the data collection”. However, such detail needs to be included in the manuscript, not in the rebuttal letter.

Furthermore, whilst it might be “reasonable to accept that their personal experience would be reflected in their interpretation of the data collection”, this needs to be made explicit in the manuscript.

We believe that we should not explicitly reveal the extent of our patient representatives’ cancer experiences. We are very grateful for their dedicated input to our whole research programme; to mark this they have both been given authorship and they are delighted to have been able to contribute so fully to this endeavour. Some clarification has been given as suggested.

5. I could not see the interview topic guide attached as an appendix - and there is no reference in the Methods section about the topic guide being in an appendix. The authors need to include the topic guide and refer to it in the manuscript.

The interview topic guide has been included with this re-submission.

RESULTS

1. I disagree with the authors that Table 1 does not need further details or changes. It would be more informative if the column on the SYMPTOM Pancreas study were removed and instead the demographics and other details of the participants were presented according to cancer diagnosis vs. non-cancer diagnosis (even if there are no major differences between the two groups).

Table 1 has been amended as suggested.

2. Overall, the new structure of the themes and sub-themes is much clearer than before. However, simply removing the word "Theme" from Tables 2, 3 and 4 does not solve the problem – what are those words and phrases meant to represent? Are they codes? Are they sub-themes? They need a heading to explain what they represent. Table 2 is not about "initial explanations" but rather about "symptom attributions".

The themes and sub-themes have been highlighted for further clarification. The Table headings have also been clarified.

Perhaps Theme 2 (Lifestyle adaptations) should have a table of illustrative quotes, like the other themes.

We do not think this will enhance the paper as there is sufficient data in the text.

DISCUSSION

1. Overall, the Discussion is well written and includes good reflection on the strengths and limitations of the study. It would be good though to clarify, were possible, those "subsequently" diagnosed with pancreatic cancer vs. those "subsequently" diagnosed with other non-cancer conditions, as I presume that the final diagnosis occurred after the interviews were conducted.

The authors do not think that this would enhance understanding as we were not able to find any differences between those diagnosed with and without cancer. Furthermore, this would make the participants more identifiable as pancreatic cancer is a rare disease.

2. A reference is needed to the wider SYMPTOM study (paragraph 2, p. 16).

3. It might be useful to define safety-netting (p.18) and what this would mean in the context of pancreatic cancer.

Both done.

VERSION 3 - REVIEW

REVIEWER	Dr Afrodita Marcu University of Surrey, UK
REVIEW RETURNED	11-Jul-2017

GENERAL COMMENTS	I would like to commend the authors on their effort to respond thoroughly to the second round of feedback. The authors have done a good job and the manuscript reads much better now as a result of
---

	the corrections made. The recruitment process and the analytic approach are much clearer now. I have just a minor comment: on page 7, I would suggest that the authors change "thematic analysis OF inductive coding" to "thematic analysis WITH inductive coding" (not in capitals, of course). I don't need to review this again - it is a simple typo to be addressed during proof-reading and editing.
--	--

VERSION 3 – AUTHOR RESPONSE

I have responded to the reviewer's minor comment regarding the wording on page 7 and have uploaded a revised version of the manuscript.